# PHOENIX (Picking up Hidden Osteoporosis Effectively during Normal CT Imaging without additional X-rays): protocol for a randomised, multicentre feasibility study

Kenneth E S Poole ,[1,2,3] Daniel D G Chappell ,[2,3] Emma Clark,[4] Jane Fleming ,[5,6] Lee Shepstone,[7] Thomas D Turmezei,[1,7,8] Adam P Wagner ,[7,9] Karen Willoughby ,[1,2] Stephen K Kaptoge[5,6]

For numbered affiliations see end of article.

**Correspondence to**
Dr Kenneth E S Poole;
kenneth.poole@nhs.net

## ABSTRACT

**Introduction** Two million out of the UK's 5 million routine diagnostic CT scans performed each year incorporate the thoracolumbar spine or pelvic region. Up to one-third reveal undiagnosed osteoporosis or vertebral fractures. We developed an intervention, Picking up Hidden Osteoporosis Effectively during Normal CT Imaging without additional X-rays ('PHOENIX'), to facilitate early detection and management of osteoporosis in people attending hospitals for CT scans.

**Methods and analysis** A multicentre, randomised, pragmatic feasibility study. From the general CT-attending population, women aged ≥65 years and men aged ≥75 years attending for CT scans are invited to participate, via a novel consent form incorporating Fracture Risk Assessment (FRAX) questions. Those at increased 10-year risk (within the amber or red zones of the UK FRAX graphical outputs for further action) are block randomised (1:1:1) to (1) PHOENIX intervention, (2) active control or (3) usual care. The PHOENIX intervention comprises (i) retrieving the CT scans using the NHS Image Exchange Portal, (ii) Mindways QCT Pro software analysis of CT hip and spine none density with CT vertebral fracture assessment, (iii) sending the participants' general practitioner (GP) a clinical report including diagnosis, necessary investigations and recommended treatment. Baseline CT scans from groups 2 and 3 are assessed with the PHOENIX intervention only at study end. Assuming 25% attrition, the study is powered to find a predicted superior osteoporosis treatment rate with PHOENIX (20%) vs 16% among patients whose GPs were sent the FRAX questionnaire only (active control) and 5% in the usual care group. Five hospitals are participating to determine feasibility. The co-primary feasibility outcome measures are (a) ability to randomise 375 patients within 10 months and (b) retention of 75% of survivors, completing their 1-year bone health outcome questionnaire. Secondary 1-year outcomes include osteoporosis/vertebral fracture identification rates and osteoporosis treatment rates. Stakeholder acceptability and economic aspects are evaluated.

## STRENGTHS AND LIMITATIONS OF THIS STUDY

⇒ First osteoporosis screening study involving patients attending radiology CT waiting areas.
⇒ Individuals randomised to either comprehensive screening for osteoporosis (by applying Fracture Risk Assessment questionnaire, CT bone densitometry and CT vertebral fracture detection) or usual care.
⇒ Allows patients the flexibility to self-consent, and tests both the willingness of CT attenders to consent, CT technologies and information flow between trial centres and Cambridge hub.
⇒ Limited by an amendment of the primary outcome measure because recruitment during COVID-19 pandemic predicted skew towards the more seriously ill (via cancellation of non-urgent and routine scans).
⇒ Modifications to both the retention outcome and longer recruitment window were made to take account of pandemic redeployment of medical research team.
⇒ Missing a secondary care direct intervention arm, which would relieve some of the burden from primary care to enact osteoporosis treatments.

**Ethics and dissemination** Approved by committee (National Research Ethics Service) East of England (EE) as REF/19/EE/0176. Dissemination will be through the Royal Osteoporosis Society (to patients and public) as well as to clinician peers via national and international bone/ rheumatology scientific and clinical meetings.
**Trial registration number** ISRCTN14722819.

## INTRODUCTION

Older patients attending hospital for CT scans may have undiagnosed spine fractures or low bone density and therefore stand to benefit from osteoporosis screening. Timely diagnosis and treatment of osteoporosis

benefits many patient outcome domains (pain, quality of life, morbidity and in the case of zoledronate treatment, mortality). An estimated 43% of patients >60 years of age undergoing CT scans (for abdominal and pelvic problems unrelated to their bones) were found to have osteoporosis or vertebral fractures when scan images were examined using computer-analysis methods.[1] However, radiologists reviewing CT scans reported only 16% of vertebral fractures in routine practice,[2] and bone mineral density (BMD) is rarely calculated from CT scans (CT BMD). This is despite CT BMD being approved internationally for the diagnosis of osteoporosis. Also, femoral neck BMD from CT is interchangeable with standard dual energy X-ray absorptiometry (DXA) BMD when using the global individual fracture risk web calculator Fracture Risk Assessment (FRAX). Indeed, several approved software packages now allow accurate diagnosis of osteoporosis in the hip and spine from routinely acquired CT images, while at the same time facilitating vertebral fractures identification and grading.

Systematic reviews indicate that overlooking osteoporosis or vertebral fractures can lead to serious, preventable pain, disability and health costs (>£2 billion per year from osteoporosis in the UK).[3] Osteoporosis is a common disease in older women and men: osteoporosis is associated with >200 000 vertebral fractures and >85 000 hip fractures annually in the UK.[4] Osteoporosis-related fractures have a high morbidity and, despite low cost-effective treatments, women and men are increasingly presenting at clinics with advanced multiple vertebral fractures, loss of body height, compressed abdominal contents, dysphagia, severe and chronic pain and reduced mobility. Despite an ageing UK population, osteoporosis remains underdiagnosed and undertreated.[5] The osteoporosis case-finding rate in primary care is low and there is no national mandate for screening. High-risk patients may be referred for DXA and are offered treatment if fracture risk is sufficiently high. Vertebral fractures are infrequently detected in usual care even though they (alongside hip fractures) cause the biggest burden to patients and the health service. We hypothesise that targeted screening will improve on usual care and improve diagnosis. The Picking up Hidden Osteoporosis Effectively during Normal CT Imaging without additional X-rays (PHOENIX) intervention is a simple and widely practicable intervention to facilitate osteoporosis identification, diagnosis and treatment in patients attending for routine clinical CT at any hospital. It has been developed to apply software to CT scan images and repurpose them for measuring bone density, fracture risk and to identify crushed vertebrae.[6 7] It couples timely diagnosis with an individualised report (built from a guideline-derived clinical decision tree) sent to general practitioners (GPs) containing advice on recommended investigations, National Institute for Health and Care Excellence (NICE)-approved, cost-effective treatment/lifestyle advice to reduce osteoporotic fractures.[8] We expect benefits to accrue through early diagnosis and treatment of

---

**Box 1    Opportunistic osteoporosis screening of CT scan attenders and rationale for Picking up Hidden Osteoporosis Effectively during Normal CT Imaging without additional X-rays (PHOENIX) intervention**

⇒ CT attenders are a high-risk population: >30% of older adults have undiagnosed vertebral fractures or osteoporosis that can be effectively treated once identified.[13]

⇒ The PHOENIX intervention diagnoses osteoporosis on scans already undertaken, so there is no requirement for additional X-ray exposure or hospital attendance.

⇒ The PHOENIX intervention can be applied to avoid the necessity of extra dual energy X-ray absorptiometry (DXA) visits and DXA reporting in multiple clinical scenarios, such as in the setting of radiology reporting quality improvement programmes that increase CT diagnosis of vertebral fractures, or to formally diagnose patients with osteoporosis after large-scale artificial intelligence (AI) vertebral fracture identification methods (eg, Optasia or ZEBRA AI1 tools).

⇒ Screening is effective at reducing hip fractures. A recent trial of primary screening based on clinical risk factors and targeted bone density scanning reduced the hip fracture incidence by 28% over 5 years, with a number needed to screen (NNS) of 111 to prevent one hip fracture.[14]

⇒ Systematic reviews of screening for osteoporosis suggest a NNS of 43 people (aged 75 years or over) to prevent one vertebral fracture.[15]

⇒ Fracture risk is currently assessed opportunistically by general practitioners. Current advances in IT software could augment this practice to make more efficient and effective use of healthcare resources, as recommended by the National Institute for Health and Care Excellence, Clinical Guideline 146.[15]

⇒ A 2013 study found that radiologists failed to report 84% of clinically important vertebral fractures from routine CT scans.[4] However, once combined with new computerised analysis methods, CT scanners can effectively determine bone density and fractures.[7]

⇒ There is a fourfold risk of another vertebral fracture occurring after the first one[16] confirming the progressive nature of established osteoporosis.

⇒ Generalisability: 1 million CT scans were performed in the UK in 1996, increasing to 5 million in 2016,[17] permitting screening of many 'at-risk' patients. Interventions such as PHOENIX are not complicated, nor technically demanding for those hospital staff who routinely analyse DXA scans.

---

osteoporosis, with additional justifications for investigating this population shown in box 1. Preventing fractures could lead to decreased societal and social burden, and reduced healthcare and social care costs.

## METHODS AND ANALYSIS

This feasibility study aims to test our PHOENIX intervention, which begins by offering adults attending hospitals for CT (for any reason) a comprehensive bone health screen, followed by 'opportunistic' reuse of their CT scans for bone densitometry. The intervention ends with their GPs receiving detailed written advice on managing the participants' bone health. Patients are offered an invitation pack comprising a novel informed consent form/data capture tool that the patient can fill out with or without assistance, and a participant information sheet.

Specifically, we want to determine whether it is feasible for our team to undertake a large trial of the PHOENIX intervention to prevent fractures in a larger population of CT attenders. Hence, our co-primary outcome measure is our ability to randomise 375 'high-risk' participants within 10 months. For the larger fracture prevention trial to be feasible, the other co-primary outcome measure must be achieved, which is retention at 12 months; our ability to record bone active treatment data (yes/no) in 75% of surviving patients. Power calculations for any future trial will be informed by the secondary outcome measure; the percentage of participants in each arm who have (i) received a treatment recommendation (due to osteoporosis, vertebral fracture or high FRAX risk) and (ii) received and commenced treatment 12 months after their CT scan. Box 2 contains a summary of the key features of the study protocol.

The PHOENIX study is a parallel three-group, non-blinded randomised controlled study of volunteers aged ≥65–90 years (women) or ≥75–90 years (men) attending for CT for any clinical reason, where the spine and/or hips are visible in scan images. Specifically, CT-attending patient volunteers who have an elevated 10-year fracture risk (within the amber or red zones of the UK FRAX graphical outputs for further action) are randomised to one of three groups. The three arms allow comparison between (1) the PHOENIX intervention, (2) an alternative active control 'middle' group (to assess whether GPs will investigate or treat patients based solely on receipt of a populated FRAX questionnaire) and (3) a control ('usual care') group where no action is taken. Potential participants are identified ahead of attendance by the clinical or research team reviewing CT lists. Those eligible by age, sex and region of the body being scanned are handed the PHOENIX pack while attending the CT department. The specially designed informed consent form (online supplemental file 1) first guides the potential participant through eligibility screening questions (*Do you take medication for your bones once a week that requires you to sit or stand upright? Do you have a yearly infusion (drip) or injection for your bones? Do you have a hip replacement? Have you ever had a hip operation?*). If there is an affirmative answer to any of these questions, individuals are asked to return their forms and go no further. Those individuals who answer 'no' to all the questions are asked to supply key demographic data: date of birth, height, weight, country of origin and ethnicity (needed for accurate FRAX calculation) and National Health Service number (receptionists support the patient as needed with the latter). Individuals are also asked to document the reason for their scan, if they know it.

On the last page of the consent form, the eight validated bone health-relevant questions from the FRAX questionnaire are completed, along with patient address, telephone number, GP name and GP address. Completed paperwork is scanned and sent by email attachment within the national, clinically secure NHS.net email account infrastructure to our add-tr.p*******@nhs.net

## Box 2  Summary of key features of the Picking up Hidden Osteoporosis Effectively during Normal CT Imaging without additional X-rays (PHOENIX) study protocol

### Pragmatic features
⇒ Patients attending CT waiting rooms are offered a PHOENIX invitation pack comprising a novel informed consent form/data capture tool to ascertain Fracture Risk Assessment (FRAX) score (online supplemental file 1) that the patient can fill out with or without assistance
⇒ The intervention (tested in group 1) comprises (i) retrieving the CT scans of those 'amber' or 'red' risk individuals with moderate to high 10-year fracture risk by FRAX, (ii) hip and spine bone mineral density analysis with vertebral fracture assessment and (iii) sending the general practitioner (GP) guidance from a clinical decision tree.
⇒ The intervention take place 'behind the scenes' without impacting on patient flow or the radiography/radiology workflow.
⇒ Data (imaging, clinical risk factor, Patient Identifiable Information (PID) including sensitive clinical information) flow via secure encrypted and approved methods including the NHS Image Exchange Portal, Picture Archiving and Communications System (PACS) and nhs.net secure clinical email.
⇒ Technical staff can 'pull' clinical CT images for assessment at any time.

### Primary outcome measures/end points
⇒ Feasibility of recruitment: defined as ability to randomise 375 'at-risk' participants over continuous 10-month period.
⇒ Retention: defined as retaining 75% of surviving participants to 12 months postrandomisation, proven by receipt of follow-up postal questionnaire (specifically an answer to the bone active medication received; yes/no question).

### Secondary outcome measures/end points
⇒ New vertebral fracture identification across three arms assessed at 12 months.
⇒ Osteoporosis identification across three arms assessed at 12 months.
⇒ Bone active medication initiation across the three arms assessed at 12 months.
⇒ Survival at 12 months across three arms assessed at 12 months.

### Inclusion criteria: participants eligible for the trial should fulfil all of the following at randomisation:
⇒ Attending for a routine clinical CT scan.
⇒ Able to provide informed consent.
⇒ Aged ≥65–90 years (women) or ≥75–90 years (men) attending for CT for any clinical reason, where the spine and/or hips are visible in scan images.

### Exclusion criteria
⇒ Metalwork in hips (due to streak artefact on the contralateral hip within CT).
⇒ Known to be receiving prescription treatment for osteoporosis other than calcium/vitamin D (ie, bisphosphonate drug, strontium ranelate, denosumab, raloxifene or teriparatide since it would be futile to identify already known and treated osteoporosis).
⇒ Prone CT scan (which cannot currently be processed by the software).

### Role of patients and the public
⇒ Actively participated in the study design in focus groups within the project team.

Continued

## Box 2 Continued

⇒ Assisted in the design of data collection tools, including the informed consent form and information leaflet.

⇒ Provided guidance on the acceptability of approaching participants in CT scan areas and insight into the anxiety of attending hospital among some more elderly members of society. Adding country of origin and ethnicity questions to consent form to facilitate FRAX.

Impact of COVID-19 pandemic

⇒ Recruitment had to be paused due to the COVID-19 pandemic, then restarted in just one centre without face-to-face contact with patients. Therefore, recruitment was either by clinical approach/discussion versus self-consent based on just handing out recruitment packs after COVID-19 restrictions were brought in.

Note that a limitation of our methodology is that the protocol was designed before the pandemic when our standard practice (enacted by fracture liaison service and rheumatologists/endocrinologists/orthogeriatricians) was to ask the GP to commence prescriptions for osteoporosis in the patients that we identify. We have witnessed the pressure and difficulties faced by primary care during the pandemic leading us to consider how we would redesign the study to reflect current healthcare pressures and primary care circumstances. Hence, if it were possible, we would now design the study with an extra arm testing the efficacy of treatments organised by the FLS team or secondary care (rheumatologists/endocrinologists/orthogeriatricians), of course informing the primary care physician of the decision to offer treatment.

address (except for local patients where the paperwork is collected directly from the CT reception). FRAX answers are processed by the research team at Cambridge who use the results to calculate the 10-year fracture risk score *without* femoral neck BMD online. Those at low risk of fracture (within the green zone of the UK FRAX graphical outputs) are not randomised, but for completeness, these low-risk participants do eventually have the bone parameters in their CT scans reviewed. CT scans from participants in this 'green' category are retrieved later and put through the PHOENIX intervention 13 months after they consented, allowing the study team to ascertain whether FRAX had been effective in picking up patients with osteoporosis/high fracture risk in this CT waiting room setting. Those in the remaining 'red' and 'amber' risk categories are randomised to one of three arms, described below: group (1) the PHOENIX intervention, (2) active control (FRAX questionnaire results only sent to GP) and (3) usual care. The flow of patients through the study, including randomisation is shown in figure 1.

### Group 1

The PHOENIX intervention. In brief, bone health for the red/amber risk participant is assessed using the CT images already captured, with computer-aided diagnosis of vertebral fractures performed by one of two trained NHS and accredited bone density technicians using Mindways QCT Pro (Mindways, Austin, Texas, USA) in Cambridge. The output is a consultant clinician-verified report (figure 2) sent to the GP with details of (i) femoral neck and/or spine bone density results, (ii) vertebral fracture diagnosis (yes/no/level affected), (iii) FRAX estimated 10-year risk of major osteoporotic fracture, (iv) any

recommended follow-up investigations and a (v) simple treat/do not treat decision based on National Osteoporosis Guideline Group (NOGG) intervention thresholds, with a suggestion of specific treatments recommended by NICE guidelines (the decision tree is shown in online supplemental file 2). These steps can take place at any time after the patient has gone home from having their CT. Any CT scanner to be used for CT densitometry must be calibrated by reference to a QCT phantom and recalibrated in the event of any X-ray tube change. The technical aspects of the PHOENIX intervention done at study entry in group 1 (and at study exit in the others two groups) are as follows: all analysis is conducted within the Mindways QCT Pro software tool using one of two stand-alone hospital-connected PCs as illustrated by screenshots (figure 3). Technical steps involved in the full PHOENIX intervention are mentioned in box 3.

### Group 2

Active control. Only the completed FRAX questionnaire results are sent to the GP, who can follow NOGG guidance themselves by entering responses onto the online FRAX website calculator. However, the patient's data are put through the PHOENIX intervention at study exit (*dilligence phase* at 13 months) and results shared with the GP then to enable them to make informed decisions regarding possible treatment.

### Group 3

Usual care. The GP receives a letter informing them of their patient's involvement in the study, but no further reports or information at this point. However, the patient's data are put through the PHOENIX intervention at study exit (*dilligence phase* at 13 months) and results shared with the GP then to enable them to make informed decisions regarding possible treatment.

Baseline and 1-year follow-up postal questionnaires are sent to all randomised participants to collect data on quality of life and resource use relevant to bone health for health economic analysis. For non-responders, a telephone interview is attempted aiming to collect the questionnaire data. Patient and clinician/healthcare worker/administrator perspectives of study involvement are explored through semi-structured interviews by telephone. All study data are held securely processed in accordance with the Data Protection Act 2018 and not disclosed to third parties. Manual records will be held securely (eg, in locked filing cabinets). Electronic records will be held on a secure network requiring user ID and password access. Individuals are not identifiable from the REDCAP and JMP (V.15, SAS Institute) spreadsheet results of the trial. The Project Management Team members (KW, KESP, DC, JMB) are responsible for the day-to-day running of the study to discuss practical aspects and ensure it is progressing. The Trial Management Group (TMG) comprises the protocol authors and one patient and public involvement (PPI) member. They are responsible for the overall conduct of the study and ensuring the

Clinical staff identify those attending for abdominal/pelvic CT scans within PHOENIX study age range

Patient attends for CT. Participant information & PHOENIX form given to patient by receptionist/ usual care team.
*A single folded A3 sheet guiding pt. through pre-screening questions, FRAX questionnaire, key personal identifying information & self- consent section*

Patient does not consent ← Patient reviews PHOENIX form & information → Patient not eligible

Patient completes form; pre-screening questions, FRAX questions incl. height, weight, reason for scan, country of origin, ethnicity, address, postcode, tel. no., GP details, NHS no. (with help from receptionist), consent boxes.

**Patients with low 10-year risk by FRAX (Green*)** Not randomised to study.
*\*National Osteoporosis Guideline MAJOR or HIP https://tinyurl.com/4au8uvut*

Research team collects form & scans it into secure NHS.net ⇒ emailed to Cambridge Hub PHOENIX nhs.net account where FRAX risk score calculated.

**Patients with higher 10-year risk by FRAX (Red/Amber*)**
FRAX score entered into REDCAP database, triggers **RANDOMISATION** n=375 participants
*stratified by centre, sex, advanced age (females≥75yrs, male ≥80yrs).*
*\*National Osteoporosis Guideline MAJOR or HIP https://tinyurl.com/4au8uvut*

| Group 1, n=125 **PHOENIX (FRAX, CT-BMD & VFA)** | Group 2, n=125 **Active Control (FRAX form only)** | Group 3, n=125 **Usual Care** |
|---|---|---|
| • CT scan retrieved, reviewed by technician<br>• Bone Density (CT-BMD) calculated & any vertebral fractures identified<br>• Clinical report sent to GP with consultant-approved management/ treatment recommendations based on National Osteoporosis Guidelines (incl. FRAX with BMD score recalculated) | • Completed FRAX questionnaire (*but not calculated FRAX score*) sent to GP who can calculate score and decide if any investigation or treatment is necessary<br>• No retrieval or review of CT scans by Cambridge Hub team **until after Month 12 (diligence phase)** | • GP notified of patient participation<br>• No other information sent to GP until study end<br>• No retrieval or review of CT scans by Cambridge Hub team **until after Month 12 (diligence phase)** |

***Immediately post-randomisation;*** short health questionnaire posted to patient- Bone medications, MSK service use, Quality of Life, Patient views. Telephone call to assist if not returned within month *(pre-contact mortality check via NHS SPINE)*

***Month 12 (final). Primary & Secondary Outcomes;*** via short health questionnaire posted to patient – Bone medications *(key outcome)*, MSK service use, Quality of Life, Patient views. Telephone call to assist if not returned within month *(pre-contact mortality check via NHS SPINE)*

***After Month 12 (diligence phase)*** Cambridge Hub team calculates FRAX scores for all remaining participants in Groups 2, 3, and Green (low risk), Bone Density (CT-BMD) calculated & any vertebral fractures identified. Report sent to GP with consultant-approved management/ treatment recommendations based on National Osteoporosis Guidelines (incl. FRAX with BMD score recalculated)
*(pre-contact mortality check via NHS SPINE)*

**Figure 1**    Study flow chart for the Picking up Hidden Osteoporosis Effectively during Normal CT Imaging without additional X-rays (PHOENIX) study. BMD, bone mineral density; GP, general practitioner; FRAX, Fracture Risk Assessment; VFA, vertebral fracture assessment; MSK, Musculoskeletal.

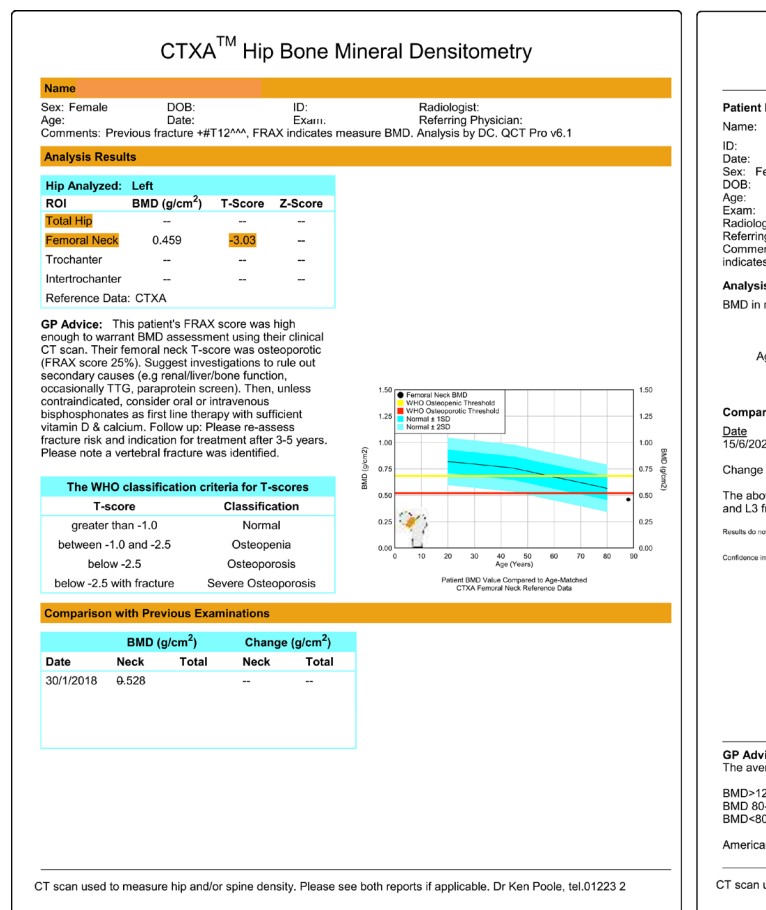

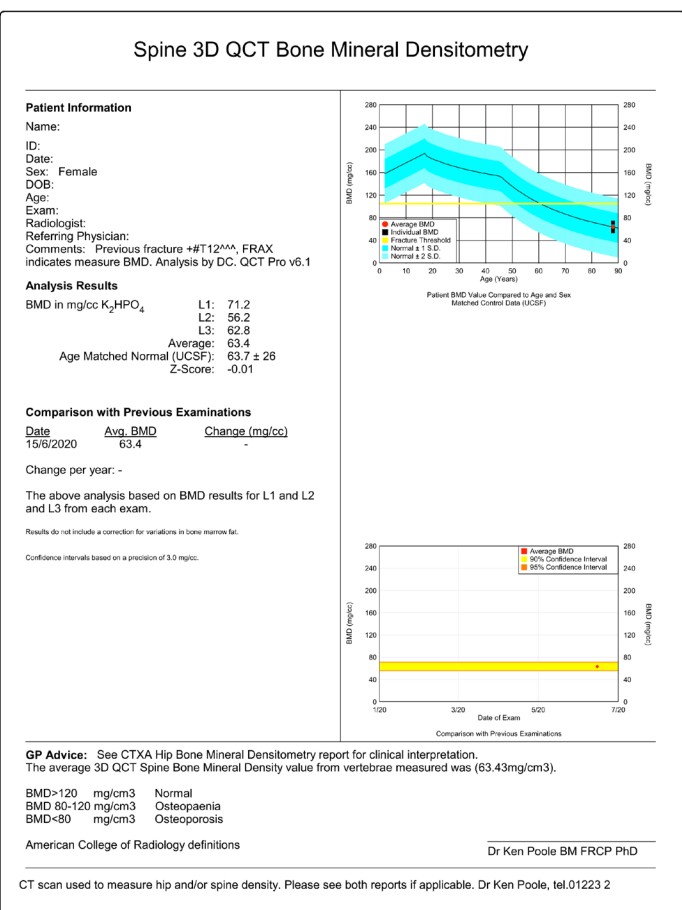

**Figure 2** Written clinical guidance and report produced at the end of a Picking up Hidden Osteoporosis Effectively during Normal CT Imaging without additional X-rays (PHOENIX) intervention in a typical patient from group 1. These comprise the standard output pdfs of Mindways QCT Pro which are sent to the participants' general practitioner (GP). In this case, there are both hip and spine results from a typical patient with a previously undiagnosed grade 3 vertebral fracture of T12 (found and graded using SlicePick). The femoral neck bone mineral density (BMD) T-score is –3.03 and spinal osteoporosis is shown with L1-L3 volumetric BMD of 63.4 mg/cm³. Their Fracture Risk Assessment (FRAX) 10-year risk of fracture is 25% and investigation/treatment is recommended by NOGG. The written clinical guidance is derived from the PHOENIX clinical decision tree (online supplemental file 2).

National Institute for Health and Care Research (NIHR) milestones are achieved. Prior to COVID-19, separate PPI meetings were held which enabled those attending to influence the study design.

## Sample size calculations, study power, randomisation method

Our primary intention is to evaluate the technical procedures needed for a definitive trial with fracture incidence (rather than recruitment/retention) as the primary outcome. To help set the target recruitment numbers for the feasibility study, we therefore used the surrogate outcome of osteoporosis treatment rates at 12 months follow-up to permit sufficient power to assess and conduct cross-group comparisons across trial arms within the period of recruitment to the feasibility study. Scenarios for scaling up to a definitive trial are based on previous data.[1 2 9] Only those participants in the red/amber FRAX group (figure 4A) will be randomised, so the number of invitations needed to reach any recruitment target will also depend on the prevalence of the red/amber FRAX group. In our local preliminary application of

the PHOENIX intervention in 1518 patients, the prevalence of red/amber FRAX varied between 30% at age 40 years and 50% at age 95 years (figure 4B), with overall prevalence 40.5% (95% CI 35.7% to 45.5%). There is no published UK data to compare with this but comparable prevalence was reported in a single study in the USA investigating combined bone density and fracture detection in routine CT investigations, where 42.3% (of 571 consecutive patients) were newly detected as having osteoporosis.[1] A conservative assumption from our preliminary work is that the PHOENIX intervention will identify a further 9.6% of screened patients for treatment and that GPs will act to treat at least half of these, resulting in an expected 4.8% minimum detectable difference. If, in our study, the percentage treated at 12 months in group 2 (active control FRAX-GP arm) matches that observed in the MRC-funded osteoporosis screening Screening in the community to reduce fractures in older people (SCOOP) study[6] (15.5%), then we would consider that an improvement in percentage treated of 4.8% would be

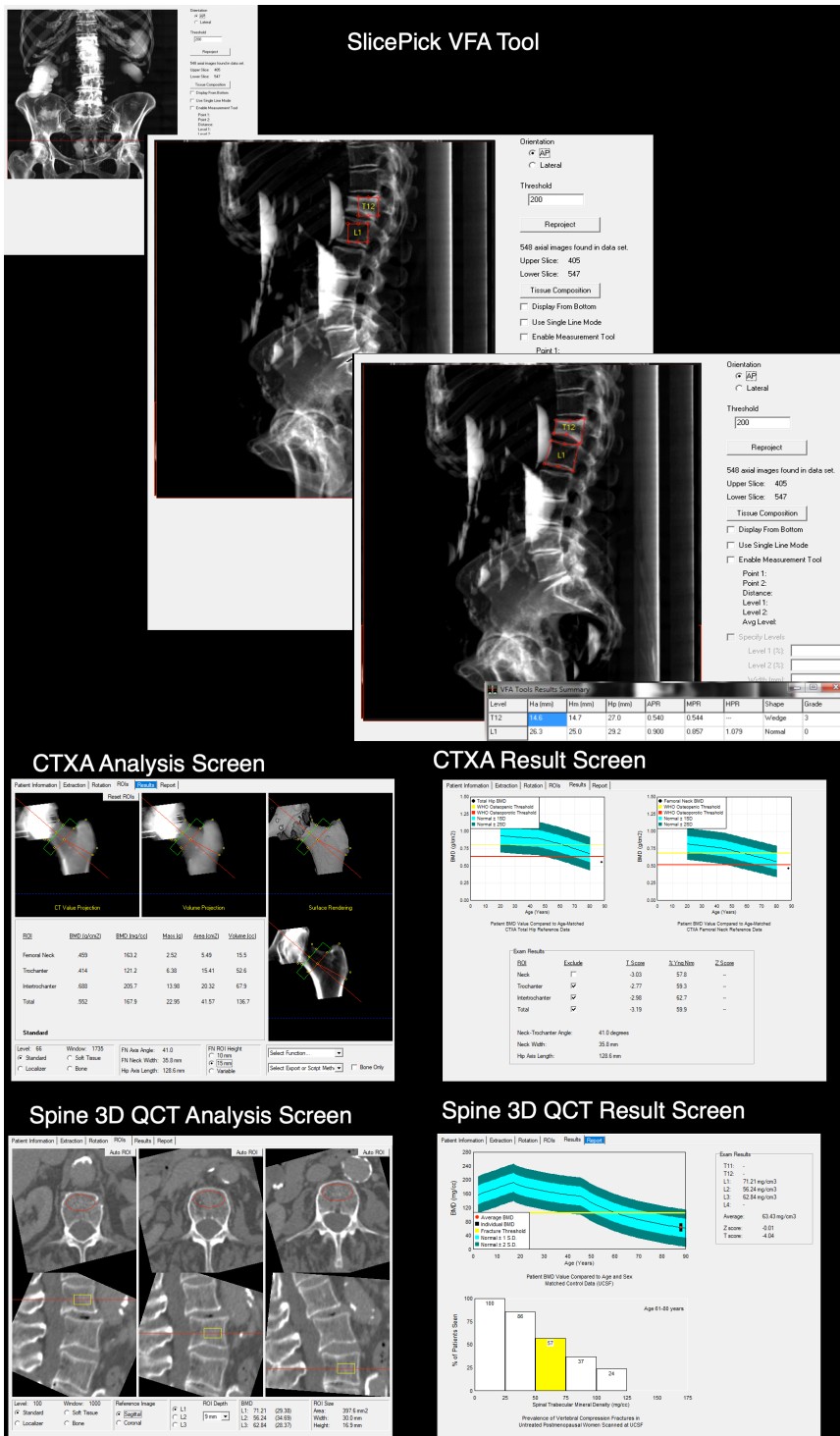

**Figure 3** The *Picking up Hidden Osteoporosis Effectively during Normal CT Imaging without additional X-rays (PHOENIX)* intervention*:* technical steps involved in vertebral fracture assessment using Mindways SlicePick to automatically create the sagittal synchronised image in a patient where the technician has used embedded 6-point morphometry to grade a previously undiagnosed T12 vertebral fracture (upper panels). Note the radiopaque bowel contrast in anteroposterior and lateral SlicePick views. SlicePick is then used to create a volume for femoral neck (CTXA) bone mineral density (BMD) and spine volumetric BMD measurements from the same CT scan. The middle panels show screenshots from CTXA femoral neck areal BMD (g/cm$^2$) analysis from CliniQCT. Note that in this case, only the femoral neck BMD value is taken forward to the report/guidance (figure 2), since the CT scan did not cover sufficiently far below the lesser trochanter to allow for 'total hip' BMD measurement. The lower panels show three-dimensional (3D) QCT analysis of lumbar spine L1, L2 and L3 trabecular volumetric BMD (mg/cm$^3$). Note that lumbar spine volumetric BMD T-scores are *not* used in the written clinical guidance and report shown in figure 2, since trabecular bone spinal vBMD T-scores are not interchangeable with dual energy X-ray absorptiometry-derived T-scores. This patients' actual pdf results file containing treatment recommendations (derived from the clinical decision tree online supplemental file 2) is shown in figure 2.

---

**Box 3    The Picking up Hidden Osteoporosis Effectively during Normal CT Imaging without additional X-rays (PHOENIX) intervention (group 1): technical and practical steps**

1. CT scan retrieved via PACS into Mindways via its DICOM transfer tool (Cambridge) or in the case of the spoke sites, via the UK NHS Image Exchange Portal to Cambridge, followed by importing to Mindways QCT Pro.
2. The Mindways QCT Pro SlicePick vertebral fracture assessment tool is used to identify any fractures in the imaged CT volume during the bone density assessment, by applying 6-point morphometry criteria to the synthesised lateral image (upper panels of figure 3). SlicePick works by synthesising anteroposterior and sagittal images from the axial CT slices.
3. Mindways QCT Pro CTXA module is used to measure bone mineral density (BMD) in the left femoral neck (figure 3, middle panes), left total hip region (if coverage is sufficient) and/or lumbar spine (L1-L3 as per International Society for Clinical Densitometry guidelines, lower panes, figure 3).
4. Each individual's Fracture Risk Assessment (FRAX) fracture risk estimate is then recalculated online, using clinical Risk Factors (RFs) from FRAX and the femoral neck BMD value. Importantly, the 'previous fracture' category is switched to 'yes' in FRAX if spine imaging revealed any vertebral fracture.
5. The NOGG guidance tool button within FRAX is then used and the pop-up red/green risk charts followed to determine whether treatment is recommended, provided no contraindications exist. In the case of no femoral neck BMD possible (spinal BMD only), FRAX values are reported (without BMD input) and the PHOENIX clinical decision tree for spine BMD is followed to produce the report that is sent to general practitioners (GPs).
6. The clinician-verified report phrases are derived from the PHOENIX clinical decision tree (online supplemental material 2), which is stored within the QCT Pro reporting tool and updated regularly. The report phrases appear (figure 2) on the Hip pdf (CTXA Hip Bone Mineral Densitometry, 'GP Advice') analysis and the Spine pdf (Spine 3D QCT Bone Mineral Densitometry 'GP Advice'). The technicians can select the correct GP advice phrase via drop-down menus when they have the FRAX and BMD results to hand. By selecting the matching patient category in the software, a bespoke template is imported into the pdf of the bone density report incorporating all the measured bone health information and clinical risk factors, based on the National Osteoporosis Guideline recommendations.
7. The final Hip and Spine pdf is signed off electronically by the consultant physician (KESP) via nhs.net email (secure clinical, encrypted) and posted to GPs with a cover letter.
8. Any vertebral fractures overlooked by radiologists on the Radiology Information System (RIS) report are entered in the local radiology discrepancy review process and the original CT report addended accordingly.
9. Responsibility for subsequent investigations including referrals for dual energy X-ray absorptiometry and/or treatment, remain with the GP.
10. A small, cylindrical Mindways model 4 (asynchronous CT calibration phantom) must be scanned at intervals on each CT scanner, including all the various kilovoltage peak energies used clinically, according to a Standard Operating Procedure. This phantom can be posted or taken to radiology departments for quality assurance scanning.

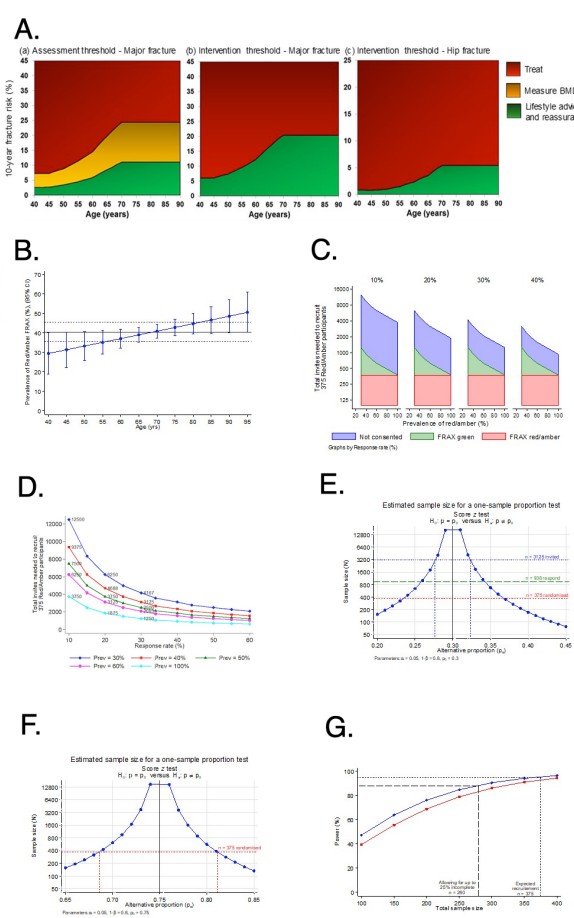

**Figure 4**    Fracture Risk Assessment (FRAX), sample size and power calculations for Picking up Hidden Osteoporosis Effectively during Normal CT Imaging without additional X-rays (PHOENIX). (A) FRAX UK age-specific fracture risk assessment and intervention thresholds. (B) Age-specific prevalence of red/amber FRAX risk category from preliminary local application of PHOENIX intervention in Cambridge. (C) Calculation of the number of invites needed to randomise 375 red/amber risk participants based on different responses (rates of agreeing to participate). (D) Number of PHOENIX invitation packs that must be given out in order to randomise 375 red/amber risk participants based on different prevalence of red/amber FRAX among participants. (E) Our feasibility study aiming to randomise 375 of 938 (40%) consenting participants (ie, 30% of 3125 total invited) would estimate the trial's response rate with a precision of 1.6% (ie, half-width of the 95% CI, *this sample size having 80% power to reject null hypothesis at a 30% response*). (F) The randomised sample of 375 patients will provide 80% power to detect a difference of at least 6% or larger in the 12-month overall trial retention rates from a hypothesised 75% retention, based on two-sided one-sample test of proportion at 5% significance level (ie, providing statistical evidence that the overall retention may be as low or high as 69% (0.69) or 81% (0.81) respectively, as opposed to 75%, 0.75=*sample size for 80% power to reject null hypothesis of 75% retention*). (G) Power versus sample size to detect differences in 12-month treatment rates in our three-arm trial, assuming treatment percentages of 4.5% vs 15.5% vs 20.3% with 1:1:1 randomisation to usual care versus active control FRAX-GP versus PHOENIX intervention, respectively, and type 1 error=0.05.

clinically significant. Figure 4C shows the estimated total number of invitees needed to recruit the 375 red/amber risk participants, assuming different response rates and different prevalence estimates. In the scenario of 30% response (similar to SCOOP) combined with 40% prevalence of red/amber FRAX (red line in figure 4D), it would be necessary to invite a total of 3125 participants (or 625 invites per centre) to achieve the recruitment target of 375 randomised participants (75 per centre, figure 4D).

## Power for recruitment and retention
Hypothesising that the recruitment (ie, consent, FRAX completion, eligibility and CT imaging, ie, 'response') would be similar to that observed in the SCOOP primary prevention trial (p0=30%); a feasibility study aiming to randomise 375 of 938 (40%) consenting participants (ie, 30% of 3125 total invited) would estimate the trial's response rate with a precision of 1.6% (ie, half-width of the 95% CI, figure 4E). The randomised sample of 375 patients will provide 80% power to detect a difference of at least 6% or larger in the 12-month overall trial retention rates from a hypothesised 75% retention (as with the SCOOP trial) based on two-sided one-sample test of proportion at 5% significance level (ie, providing statistical evidence that the overall retention may be as low or high as 69% (0.69) or 81% (0.81), respectively, as opposed to 75%, 0.75, figure 4F). In addition, against the same expectation, the 1:1:1 randomised comparison (ie, PHOENIX intervention vs active control FRAX to GP vs usual care pathways) will have >80% power to detect at least 12.2% absolute difference in retention rates between trial arms (or RR=1.16) based on two-sided two-sample test of proportion at 5% significance level.

## Power for treatment proportions
Assuming that osteoporosis treatment percentages at 1-year follow-up in the usual care versus active control FRAX-GP versus PHOENIX intervention would be 4.5% vs 15.5% vs 20.3%, respectively (ie, as observed in SCOOP trial vs with expected improvement in PHOENIX), a total of 375 participants recruited and randomised in the ratio 1:1:1 will have 96% power (figure 4G) to detect linear trend in treatment proportions based on a two-sided test of binomial proportions at 5% statistical significance level. The same sample size will have 93% power for a secondary two-degree of freedom test of between-group contrasts in comparison with usual care. When allowing for up to 25% attrition due to non-differential dropout or incompleteness of outcomes, the remaining sample size of 280 participants will have >84% power to detect the same magnitude of differences (figure 4G).

## Randomisation method
Knowing that 375 participants were needed, we were advised to undertake block randomisation stratified by centre, sex and age group (female: <75 vs ≥75 years, male: <80 vs ≥80 years). Randomisation uses the web application

Research Electronic Data Capture (REDCap https://www.project-redcap.org/).

## Statistical analysis
Daily screening logs will be kept determining: (i) the total number of days (n) where patients were approached in each centre and (ii) the daily totals of PHOENIX invitation packs distributed. This log will also record if a numbered pack was not given out on a day (eg, if the patient did not arrive for scanning, or the receptionists omitted to give out the pack). The total number of consented individuals (x) as well as the number of consented individuals per day (total x/n overall, and per centre) will be calculated. The number of consented but subsequently excluded individuals (y) will be kept (eg, for subsequently discovered metalwork in the scan, technical issues with scan). The number of randomised participants will be calculated (z=x–y), and the rate of randomisation (z/n overall, and per centre). These data will be captured from screening logs and entered manually onto a database. The REDCap database will be used to record details of all consenting patients (z). The database will be locked on completion of data entry and when missing data or any discrepancies are resolved. The data will be exported for statistical analysis to the study statisticians with assistance from the chief investigator.

## Predefined statistical analysis plan
Response rates will be calculated as the proportion of participants invited who (a) consent to take part, (b) meet eligibility criteria and (c) are randomised, with 95% CIs to assess the extent to which the realised response deviates from the expected as detailed in the power calculations above. Cumulative participant accrual over time will be plotted overall and by recruitment centre to assess any major differences in accrual and assess progress towards target. Logistic regression will be used to compare randomised groups with respect to the 12-month outcomes of treatment rates and participant retention. Baseline stratification covariates (ie, centre, sex and age) will be adjusted for in the models. Analyses will be on an intention-to-treat basis.

## Qualitative analysis and stakeholder acceptability research
Both the baseline and 1-year follow-up questionnaires being sent to all patients include space for participants to feedback their experience of the study. First explaining how this will help, this section includes example open questions such as *"Do you have any comments on being offered a bone health check at the same time as your CT scan? Or on taking part in this research study?/how you were asked to join?/ your decision to take part?"*. Semi-structured interviews will be used with a smaller subsample of study participants (n=20) to allow for more in-depth exploration of issues that participants may raise in written feedback or during interview. Recordings will be transcribed by an academic transcription service following a confidentiality contract and anonymised by a member of the research team (JF)

changing names of people and places and any identifiable details. Key research questions for the PHOENIX process evaluation are: (i) what are the implementation facilitators and barriers for the PHOENIX intervention? (ii) how acceptable to the intended patient group is the process of PHOENIX screening? (iii) how do implementation and acceptability issues affect uptake of the CT bone health screening and any subsequent GP investigation and recommended bone protective treatment? A combination of interviews and focus groups with those staff groups who will be key to the effective delivery of PHOENIX screening will also be undertaken, both in hospital and primary care, in order to understand their different perspectives. These staff groups are research nurses, receptionists and radiographers in CT departments and GPs. Analyses will be descriptive. The software programme NVivo V.12 (QSR) will be used to manage and index the qualitative data (anonymised transcripts of audio-recordings from interviews and focus groups) before charting, mapping and interpretation taking a framework approach.[10] Ritchie and Spencer thematic analysis is an approach suitable for qualitative research in which data collection is relatively focused on key questions,[11 12] hence it is applicable to PHOENIX.

## Patient and public involvement

At the 'idea' stage of the research, our small team presented our idea at the large workshop *"Find your voice; Demystifying research and public involvement in research"* at Cambridge Professional Development Centre Trumpington in May 2015. This was run by Dr Doreen Tembo Patient and Public Involvement and Engagement (PPIE) lead for the Eastern region NIHR Research Design Service in partnership with the Norfolk and Suffolk PPIRes team (PPI in research). This connected us with patients and public representatives, many of whom had relevant musculoskeletal disorders and/or past experience in an orthopaedic service. We collated the opinions and suggestions on our proposed project from every participant in the workshop in writing, individually and from small group discussions to help develop our research question and outcome measures, so that they were informed by patients' priorities, experiences and preferences. Three individuals, Mr Jeremy Dearling, Mrs Tessa Plume and Dr Ann Frost volunteered to join our trial group as PPI representatives; they were specifically involved in patient documentation design (particularly the PHOENIX pack, informed consent form which facilitated consent without having a researcher present) and contributed to suggestions for increasing patient recruitment and follow-up. JF facilitated group sessions and individual structured interviews to assess the burden of the PHOENIX intervention on patients themselves, as well as a qualitative assessment of the process from a user perspective. The results of the research will be disseminated to surviving participants after generating patient-friendly materials in collaboration with our PPIE representatives.

## Economic analysis

Completion rates on the quality-of-life measure (EQ-5D-5L) and resource use items included within the postal survey will inform whether an economic evaluation is feasible using these data collection approaches in the definitive trial. Patterns of response and non-response will be explored to see if they suggest ways the postal survey can be refined to improve data completeness.

## ETHICS AND DISSEMINATION

Approved by committee (National Research Ethics Service) East of England (EE) as REF/19/EE/0176. Dissemination will be through the Royal Osteoporosis Society Bone Academy (to patients and public) as well as to clinician peers via national and international bone/rheumatology scientific and clinical meetings.

**Author affiliations**
[1]Department of Medicine, University of Cambridge, Cambridge, UK
[2]NIHR Cambridge Biomedical Research Centre, Cambridge, UK
[3]Rheumatology, Cambridge University Hospitals NHS Foundation Trust, Cambridge, UK
[4]Clinical Science at North Bristol, University of Bristol, Bristol, UK
[5]Cambridge Public Health, University of Cambridge, Cambridge, UK
[6]Department of Public Health and Primary Care, University of Cambridge, Cambridge, UK
[7]Norwich Medical School, University of East Anglia, Norwich, UK
[8]Radiology, Norfolk and Norwich University Hospital NHS Trust, Norwich, UK
[9]NIHR Collaboration for Leadership in Applied Health Research & Care (CLAHRC) East of England, Cambridge, UK

**Acknowledgements** The PHOENIX study is dedicated to the memory of J. Keenan Brown, inventor of QCT Pro (9 February 1960 to 11 December 2021). Without Keenan, this study would not have been possible. The authors would like to acknowledge Karen Blesic for providing a valuable nursing perspective, for helping develop the idea and for her major contributions towards the NHS Research Innovation Fund pilot to PHOENIX (the clinical service called CORTEX). The authors acknowledge the contributions of Polly Barnes and Charlotte Tyson to CORTEX and also their contributions towards the practicalities of imaging data flow. The authors acknowledge Judith Brown's contributions towards the practicalities of patient flow and analysis through the pathways and imaging services. The authors acknowledge Jeremy Dearling as our primary PPIE representative who gave valuable contributions to the design of the overall study, provided detailed feedback on the materials and protocol development. The authors also acknowledge the assistance of the Eastern region NIHR Research Design Service team, specifically Jonathan Scales, Andrew Sharpe, Analisa Casarin (improving and developing the research idea and application) and Doreen Tembo (PPI lead for the same). KESP acknowledges the Cambridge NIHR Biomedical Research Centre who support his work, and support DC. KESP also thanks the NIHR for funding his attendance at the Pragmatic Clinical Trials Course at Queen Mary University of London which helped him develop the idea. Finally, the authors acknowledge the major contribution of Dr Alan Lamont, chair of East of England (EE), Research Ethic Committee whose helpful suggestions and understanding of the relevant GCP issues and legislation rekindled our interest in adopting pragmatic patient consent without a researcher present.

**Contributors** KESP: chief investigator for the study and is responsible for the concept and design of the overall study and manuscript final approval. KESP and EC provided clinical expertise in the area of bone health which forms the basis for the study. They developed the concept and idea for PHOENIX, gained funding and wrote the clinical decision tree with assistance from DC. Also methods design, writing and providing feedback of manuscript, manuscript final approval. KESP is responsible for the medical aspects of opportunistic diagnosis of osteoporosis, recommending investigations and treatments to participants. SK (with assistance from the relevant trials work of LS, and pilot work of KESP/DC) developed the statistical analysis plan including calculation of sample size and definition of primary and secondary outcomes and power calculations. APW designed the

health economics work, and developed suitable outcomes for the trial. DC provided information on the background and rationale for using the various software. He has helped clarify pathways for scan analysis in the study and assisted in the development of the Clinical Decision Tree. KW wrote the protocol and drafted the manuscript, clarified pathways and study flow and liaised with the team to finalise the protocol. JF designed the process evaluation substudy, with all qualitative interview work, and developed suitable outcomes for the trial. TT provided information on the background and rationale for diagnosing vertebral fractures. He was responsible for the medical aspects of opportunistic diagnosis of vertebral fractures. All authors contributed to and approved the final version of the manuscript.

**Funding** This project is funded by the National Institute for Health Research (NIHR) under its Research for Patient Benefit (RfPB) Programme (Grant Reference Number PB-PG-0816-20027) and by the Cambridge NIHR Biomedical Research Centre (BRC-1215- 20014). Funding is in place to 31 March 2022.

**Disclaimer** The views expressed are those of the author(s) and not necessarily those of the NIHR or the Department of Health and Social Care.

**Competing interests** None declared.

**Patient and public involvement** Patients and/or the public were involved in the design, or conduct, or reporting, or dissemination plans of this research. Refer to the 'Methods and analysis' section for further details.

**Patient consent for publication** Not applicable.

**Provenance and peer review** Not commissioned; externally peer reviewed.

**Data availability statement** Data are available on reasonable request. Researchers involved in the trial will have access to the full dataset. Applications for access to the final trial dataset will be through the Trial Management Group (TMG). All members of the TMG will have full access to the dataset, and a controlled access model (openly available to all applicants) will be followed as set out in the MRC and NIHR guidance (https://www.methodologyhubs.mrc.ac.uk/files/7114/3682/3831/Datasharingguidance2015.pdf).

**Author note** All authors agreed to be accountable for all aspects of the work to enable questions relating to the accuracy or integrity of any part of the work to be appropriately investigated and resolved.

**ORCID iDs**
Kenneth E S Poole http://orcid.org/0000-0003-4546-7352
Daniel D G Chappell http://orcid.org/0000-0002-1958-6246
Jane Fleming http://orcid.org/0000-0002-8127-2061
Adam P Wagner http://orcid.org/0000-0002-9101-3477
Karen Willoughby http://orcid.org/0000-0002-2039-1155

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
