## [Reviewer comments · BMJ Open]

ARTICLE DETAILS

TITLE (PROVISIONAL)	PHOENIX (Picking up Hidden Osteoporosis Effectively during Normal CT Imaging without additional X-rays). Protocol for a randomised, multi-centre feasibility study
AUTHORS	Poole, Kenneth; Chappell, Daniel; Clark, Emma; Fleming, Jane; Shepstone, Lee; Turmezei, Tom; Wagner, Adam; willoughby, karen; Kaptoge, Stephen

VERSION 1 – REVIEW

REVIEWER	Kennedy, Sean McMaster University
REVIEW RETURNED	05-Aug-2021

GENERAL COMMENTS	Primary concerns center around: 1. Determining BMD from CT images. Little is shared on how this will be done and what is validity of this measurement technique compared to traditional DEXA.2. Computer assisted diagnosis of compression fractures. Again, what are software details? Internal, external validity? How has this been tested? How does this compare to radiologist gold standard?
--

REVIEWER	Giovannini, Ivan University Hospital Santa Maria della Misericordia, Rheumatology Clinic
REVIEW RETURNED	09-Aug-2021

GENERAL COMMENTS	I read with interest the study protocol for screening osteoporosis. This may improve usual care and diagnosis. However, there are some points which should be addressed to clarify the protocol. I personally found intricate to recap among the many of sub-headings of the article. The authors decided to separate introduction, rationale, reasons for screening, figure, public involvement, power for recruitment and many other sub-headings. They might consider to re-write and combine some of the sub-headings. The authors mention only briefly what this “new software programmes” allows to achieve, reporting that they could measure bone density, fracture risk and identify crushed vertebrae. Is it an automatic process that doesn’t require any operator interaction? The author could consider expanding this section. The authors decided to exclude all patients hip replacement. It seems that they excluded both patients with bilateral and mono
---

	lateral hip replacement. Moreover, the authors then assert that BMD is measured in the left total hip region. It seems confusing, considering that FRAX score is available even without hip BMD score (and the authors also performed it). This choice needs to be better addressed. Furthermore, the author decided to include only women aged over 65 and men over 75. The author stated that CT attenders are already an high-risk population, so why they decided not to include patient from the age 40 to 90 years (i.e., the range of FRAX score)? The patients are assessed the first time attending the CT department, and a second time after a year. The second assessment (after a year) needs to be better assessed. Pg 7, row 23-24, the authors reported that “BMD is measured in all participants, those not randomize to group 1 will be measured later at one year” suggest that they programmed repetition of BMD assessment after a year (that is not present in the flow chart, showing as the follow-up is only a remote check). This sentence may need some re-phrasing. This protocol has been described as a feasibility study. Thus, the treatment is demanded to GPs. It seems a limitation, because the GP may be more prone to focus on osteoporosis if the patient is reporting he’s been selected for screening. The primary aim seems lacking the point. They stated that PHOENIX-f aims to determine whether new technology can improve early detection od osteoporosis or vertebral fractures by existing CD scans, nevertheless the authors do not aim to whether this approach may act as a screening tool, or if the patients start a treatment for osteoporosis at 1 year follow-up as results of this protocol. The English is generally quite easy to understand.
--	---

REVIEWER	Narloch, Jerzy Medical University of Warsaw
REVIEW RETURNED	17-Aug-2021

GENERAL COMMENTS	Dear Authors, I congratulate you on this study, opportunistic screening for low bmd should become a standard of practice given the annual number of CTs performed and decreasing dosage of current examinations. I have however one quention regarding group 1 arm. What is the role of ct technician in reviewing the exam? Is this review a solely technical one, regarding the quality of the exam, or is it a scout for vertebral fractures in which case a radiologist should be engaged?
---

VERSION 1 – AUTHOR RESPONSE

For the Reviewers;

We thank our expert reviewers and the editor for giving us the chance to substantially rewrite the manuscript and respond to their very helpful comments.

Reviewer 1 comment :

1. Determining BMD from CT images. Little is shared on how this will be done and what is validity of this measurement technique compared to traditional DEXA.

Author response : Thank you. The way we are assessing density in PHOENIX, i.e. hip areal BMD

measured using calibrated CT, has been accepted by the International Society for Clinical Densitometry as a diagnostic technique equivalent to DXA since 2015. To clarify, we have added the sentence to the introduction, "...bone density (BMD) is rarely calculated from CT scans (CT BMD). This is despite CT BMD being approved internationally for the diagnosis of osteoporosis. Also, femoral neck BMD from CT is interchangeable with standard DXA* BMD when using the global individual fracture risk web calculator FRAX (*Dual Energy X-ray Absorptiometry). Indeed, several approved software packages now allow accurate diagnosis of osteoporosis in the hip and spine from routinely-acquired CT images, while at the same time facilitating vertebral fractures identification and grading... "

Reviewer 1 comment : 2. Computer assisted diagnosis of compression fractures. Again, what are software details? Internal, external validity? How has this been tested? How does this compare to radiologist gold standard?

Author response : We have added a detailed figure 3 to explain the pipeline for computer assisted diagnosis of vertebral fractures using the commercial tool Mindways QCT. The technique is based on standard 6 point morphometry on maximum intensity sagittal projections. Please see the new figure and technical details section of the updated methods section.

Reviewer: 2

Dr. Ivan Giovannini, University Hospital Santa Maria della Misericordia

Comments to the Author:

Reviewer 2 comment : I read with interest the study protocol for screening osteoporosis. This may improve usual care and diagnosis. However, there are some points which should be addressed to clarify the protocol. I personally found intricate to recap among the many of sub-headings of the article. The authors decided to separate introduction, rationale, reasons for screening, figure, public involvement, power for recruitment and many other sub-headings. They might consider to re-write and combine some of the sub-headings. Thank you for the valuable feedback. In response we have greatly condensed and rewritten the protocol to remove many subheadings and improve the flow of the manuscript.

The authors mention only briefly what this "new software programmes" allows to achieve, reporting that they could measure bone density, fracture risk and identify crushed vertebrae. Is it an automatic process that doesn't require any operator interaction? The author could consider expanding this section.

Author response : Thank you for the suggestion. We gave this response to reviewer 1: "We have added a detailed figure 3 to explain the pipeline for computer assisted diagnosis of vertebral fractures using the commercial tool Mindways QCT. The technique is based on standard 6 point morphometry on maximum intensity sagittal projections. Please see the new figure." Expanding the process requires an operator, although as can be seen from figure 2, some steps are automated.

Reviewer 2 comment : The authors decided to exclude all patients' hip replacement. It seems that they excluded both patients with bilateral and mono lateral hip replacement. Moreover, the authors then assert that BMD is measured in the left total hip region. It seems confusing, considering that FRAX score is available even without hip BMD score (and the authors also performed it). This choice needs to be better addressed.

Author response : Thank you. There is a streak artefact from the metalwork affecting contralateral hip. Although we have metalwork streak reduction technology, it is not yet licenced within QCT Pro for clinical use. We were concerned about hip replacement participants with metalwork in situ self-consenting who might be solely undergoing a pelvic CT, and in whom we would not be able to generate a reliable bone density result, hence the exclusion. FRAX can indeed be performed without BMD which is explained better in the new technical section.

Reviewer 2 comment : Furthermore, the author decided to include only women aged over 65 and men over 75. The author stated that CT attenders are already an high-risk population, so why they decided not to include patient from the age 40 to 90 years (i.e., the range of FRAX score)?

Author response : We adopted the UK National guideline for published in NICE CG165 which governs the age and sex appropriate for a case finding interventions in the UK.

Reviewer 2 comment :The patients are assessed the first time attending the CT department, and a second time after a year. The second assessment (after a year) needs to be better assessed. Pg 7, row 23-24, the authors reported that “BMD is measured in all participants, those not randomize to group 1 will be measured later at one year” suggest that they programmed repetition of BMD assessment after a year (that is not present in the flow chart, showing as the follow-up is only a remote check). This sentence may need some re-phrasing.

Author response : We have extensively rewritten the paper since we have caused confusion for both reviewers. The only analysis done at one year is a “rescue analysis” for those who never had a baseline analysis because of the group they randomised to (Group 2, 3 and also the ‘green’ low risk patients who weren’t randomised). To clarify; each patient only has one BMD assessment, just after consent in Group 1, or alternatively if they 'missed' that baseline BMD assessment (by being randomised to Groups 2&3) then we act on this omission 12 months later and recall their own baseline CT scan images for BMD assessment at the close of the study (as a ‘rescue’ to ensure no harm ensues from a missed osteoporosis diagnosis).

Reviewer 2 comment :This protocol has been described as a feasibility study. Thus, the treatment is demanded to GPs. It seems a limitation, because the GP may be more prone to focus on osteoporosis if the patient is reporting he’s been selected for screening.

Author response : We agree, but even having been given a short participant information sheet describing osteoporosis, we suspect that many of these patients will simply not be followed up for osteoporosis assessment and management in primary care. This matches our clinical experience. At the results stage, it may be appropriate to reference a potential limitation of heightening participant awareness of osteoporosis (referral/volunteer selection bias). We do not consider that this trial will be particularly prone to this, because the presenting patients have a lot of concerns when coming to CT, of which osteoporosis is usually only peripheral. In fact, in the study we will be able to describe the eventual diagnoses made from the CT scan, i.e. the patients’ primary illness as per the radiologists’ report. We think that our participants’ will likely assume that ‘no news is good news’. As for the GPs: our GP colleagues’ response to the Group 2 (FRAX only) and their usual care (Group 3) as well as the green (low risk) groups will give some indication of the effect.

Reviewer 2 comment :The primary aim seems lacking the point. They stated that PHOENIX-f aims to determine whether new technology can improve early detection od osteoporosis or vertebral fractures by existing CD scans, nevertheless the authors do not aim to whether this approach may act as a screening tool, or if the patients start a treatment for osteoporosis at 1 year follow-up as results of this protocol.

Author response : You are quite correct and thank you for alerting to this crucial omission. We apologise for this omission and have extensively rewritten and expanded the section on the outcomes that were in the protocol but missing from the manuscript typed version of the protocol.

The English is generally quite easy to understand.

Reviewer: 3

Dr. Jerzy Narloch, Medical University of Warsaw

Comments to the Author:

Dear Authors,

I congratulate you on this study, opportunistic screening for low bmd should become a standard of practice given the annual number of CTs performed and decreasing dosage of current examinations.

Reviewer 3 comment : I have however one question regarding group 1 arm. What is the role of ct technician in reviewing the exam? Is this review a solely technical one, regarding the quality of the exam, or is it a scout for vertebral fractures in which case a radiologist should be engaged?

Author response: Thank you. The CT technician conducts solely a technical review. Any vertebral fractures identified by the Mindways SlicePick stage in the Group 1 arm analysis must go to the radiologists for confirmation and discrepancy assessment. Please see the new Figures on the analysis method.

VERSION 2 – REVIEW

REVIEWER	Giovannini, Ivan University Hospital Santa Maria della Misericordia, Rheumatology Clinic
REVIEW RETURNED	20-Feb-2022

GENERAL COMMENTS	I read with interest the updated version of the manuscript by Kenneth et al. They propose to intercept osteoporosis in the diagnostic CT scans performed for other reasons. There are no results, as the manuscript describes study protocol. In the meantime, I hypothesize that the authors are collecting the data, even in this pandemic situation. Personally, I find confusing that the Phoenix group does not include treatment, but only suggestions to GP. If I correctly understand, in the end, is always the GP that decides whether to start treatment and what treatment. I would suggest the author include a group of patients who could be evaluated by an OP specialist for treatment management, instead of the GP. In fact, the GP is always informed (even by the enrolled patient) about the ongoing study, so he could start a treatment/focus on OP regardless of the study results.
--

REVIEWER	Narloch, Jerzy Medical University of Warsaw
REVIEW RETURNED	14-Feb-2022

GENERAL COMMENTS	I approve of the corrections
------------------------------

VERSION 2 – AUTHOR RESPONSE

Dr Giovannini has expressed one of our biggest concerns about this type of research. It is a point beautifully made. The protocol was designed before the pandemic when our standard practice (enacted by Fracture Liaison Service and Rheumatologists) was to ask the GP to commence prescriptions for osteoporosis. We have witnessed the pressure and difficulties faced by primary care during the pandemic which mean that if we could design the study again, we would certainly include

the arm described by Dr Giovannini. However, this is not possible for the current study and will need to form part of future endeavours.

VERSION 3 – REVIEW

REVIEWER	Giovannini, Ivan University Hospital Santa Maria della Misericordia, Rheumatology Clinic
REVIEW RETURNED	30-Mar-2022

GENERAL COMMENTS	I read with interest the revised version of the manuscript. The authors suggest a pilot protocol for OP detection in patients undergoing CT imaging. No results are presented, since it is a feasibility study protocol. The main expected results are based on the treatment start, but the authors do not plan to provide the treatment. As previously commented, the treatment and the responsibility for the treatment is always demanded to the GP, instead of a specialistic (rheumatological or endocrinological) evaluation. Therefore this study appears as an informative tool for the GP, instead of an interventional trial. I find no additional major comments.
---

VERSION 3 – AUTHOR RESPONSE

****Your comment****:- We note that reviewer 2 is still concerned about the methods, following on from your previous response (below) please ensure that the change in standard practice is discussed as a limitation but we do not need further revisions to be made in response to their comments.

****Our response**** -: The following limitation has been added under Table 2, Impact of COVID-19 pandemic figure legend and also (briefly) in the strengths and limitations statements at the beginning. “A limitation of our methodology is that the protocol was designed before the pandemic when our standard practice (enacted by Fracture Liaison Service and Rheumatologists/Endocrinologists/Orthogeriatricians) was to ask the GP to commence prescriptions for osteoporosis in the patients that we identify. We have witnessed the pressure and difficulties faced by primary care during the pandemic leading us to consider how we would redesign the study to reflect current healthcare pressures and primary care circumstances;. Hence if it were possible, we would now design the study with an extra arm testing the efficacy of treatments organised by the FLS team or secondary care (Rheumatologists/Endocrinologists/Orthogeriatricians), of course informing the primary care physician of the decision to offer treatment.”